# Influence of Low-Impact Development in Flood Control: A Case Study of the Febres Cordero Stormwater System of Guayaquil (Ecuador)

Fabian Quichimbo-Miguitama [1,2], David Matamoros [3], Leticia Jiménez [4] and Pablo Quichimbo-Miguitama [2,5,*]

1 Facultad de Ingeniería en Ciencias de la Tierra, Escuela Superior Politécnica del Litoral, Campus Gustavo Galindo, Guayaquil EC090112, Ecuador; flquichi@espol.edu.ec
2 Departamento de Recursos Hídricos y Ciencias Ambientales, Universidad de Cuenca, Cuenca EC010207, Ecuador
3 Facultad de Ciencias Naturales y Matemáticas, Escuela Superior Politécnica del Litoral, Campus Gustavo Galindo, Guayaquil EC090112, Ecuador; dmata@espol.edu.ec
4 Facultad de Ciencias Exactas y Naturales, Universidad Técnica Particular de Loja, San Cayetano Alto s/n, Loja EC1101608, Ecuador; lsjimenez@utpl.edu.ec
5 Facultad de Ciencias Agropecuarias, Campus Yanuncay, Universidad de Cuenca, Cuenca EC010114, Ecuador
* Correspondence: pablo.quichimbo@ucuenca.edu.ec

**Abstract:** Urban flooding is a major problem in many coastal cities. The rapidly shifting patterns of land use and demographic increase are making conventional approaches to stormwater management fail. In developing countries such as Ecuador, a lack of monitoring, financial constraints and absence of proper policies exacerbate flooding problems. This work assesses the implementation of two Low Impact Development strategies (LIDs), namely, green streets and rain barrels, as nature-based solutions to mitigate flooding problems. The use of the "Stormwater Management Model" (SWMM) helped to contrast the new approach with the current state of the drainage system, including normal and extreme scenarios. With an implementation of 1.4% (19.5 ha) of the total area with LIDs, the reduction of runoff for short events (200 min) is around 20%, and for extreme events (within 24 h) is around 19% in comparison to the conventional approach. Flooded nodes were reduced to 27% for short events, and to 4% for extreme events. The peak flooding system had a reduction to 22% for short events and 15% for extreme events. These highlights help to increase city resilience, and authorities and stakeholders should engage in climate actions to reduce flood risks complementing drainage operations with nature-based solutions. Moreover, calibrated results in this article serve to increase awareness among municipal authorities regarding the importance of maintaining flooding records to improve modelling results for decision-makings processes.

**Keywords:** green–blue strategies; green streets; rain barrels; overflow; SWMM; flooding

## 1. Introduction

Coastal populations on every continent have exploded as global trade has flowed into coastal nations through international ports, creating jobs and economic activity [1]. However, progress is constantly being challenged due to the fast increase of population density and climate change [2,3]. Moreover, the latter is producing, as a frequent event, coastal flooding with high socio-economical costs [4].

Traditionally, the way to mitigate flooding problems was through gray infrastructure (e.g., concrete revetment, concrete channels to sewer lines systems), where drainage pipe systems combine rainwater and wastewater. Modern drainage systems adopt separate paths to eliminate runoff to prevent flooding. Lastly, blue infrastructure (e.g., rainwater harvesting/rain barrels, spring water collection, floodplains, underground detention/infiltration) and green infrastructure (e.g., swales, rain gardens/bioretention cells,

green streets) are alternative approaches to addressing the problem with limited impacts on nature-LIDs [5] (i.e., reducing water pollution, irrigating parks) [6]. Both green and blue infrastructure projects utilizing sustainable approaches have been successfully applied due to the trade-off among the multiple benefits of green–blue land uses in Brazil [7], China [8], EEUU [9], and the U.K. [10]. However, their applicability is still a challenge for developing countries due to scarce resources and the absence of a holistic approach [11].

Guayaquil is the second largest city in Ecuador (354.7 km$^2$) and the main economic driver of the country [12]. Due to its low elevation (~5 m a.s.l.) and its location in a low-lying delta surrounded by the Guayas River to the east and the Estero Salado estuary to the southwest, the city is vulnerable to urban flooding. Normally, the rainy season occurs from January to April, producing 80% of the city's annual precipitation. In addition, events like ENSO (positive and negative phases) have strong effects on stormwater systems [13]. However, areas such as the Febres Cordero (FC) parish in Guayaquil not only receive intense rains, but the tidal influence that blocks outfalls from the stormwater system causes floods. The rapid urbanization, deficient planning, and scarcity of green areas exacerbate flooding problems [14].

In the interest of providing solutions to mitigate flooding problems and improve our knowledge of green–blue strategies (nature-based solutions) under limited data scenarios, this paper provides a model developed on the Storm Water Management model (SWMM) [15], which has been widely used with these particular approaches [16,17]. The outcomes of this calibrated model represent an important tool for decision-making processes that may provide crucial insights to be included in future urban planning.

To carry out the present study, the objectives were to:

(i)     determine the duration of a typical rainfall event from a temporal series of normalized events processed from official meteorological data,

(ii)    determine the intensity of extreme events associated with return periods of 2, 5, 10, 25, 50, and 100 years. A similar process was performed with tidal waves, and

(iii)   evaluate the applicability of green–blue solutions to the study area by building a model in SWMM, incorporating LIDs and comparing the results obtained by the proposed approach and the traditional design.

## 2. Materials and Methods

### 2.1. Study Area

The current study was implemented at one of the 16 urban parishes of the city of Guayaquil, namely Febres Cordero (FC), which is situated next to the Estero Salado as shown in Figure 1. This parish has a total area of 1390 ha with elevations ranging from 1 m to 5 m above sea level (m a.s.l.), as seen in Appendix A, Figure A1. FC parish is characterized by high residential land used (Table 1) [18], a large impervious area (96%) in comparison with the whole city of Guayaquil (79%), and soft slopes with an average of 3%, which represents a flat area. According to the reports made in 2018 by the Development Bank of Latin America-CAF [14], the study area has a poverty ratio of 23%, which is higher than the average for the city (17%), and a high population density (24,100 people per km$^2$) in contrast with the average of the city (12,530 people per km$^2$).

**Table 1.** Type and percentage of land uses of Febres Cordero Parish.

| Land-Uses | Area (ha) (%) |
|---|---|
| Residential Areas—impermeable soil | 755.5 (54.4) |
| Commercial Areas—impermeable soil | 37.2 (2.7) |
| Industrial Areas—impermeable soil | 0.4 (0) |
| Streets—impermeable soil | 467.3 (33.6) |
| Open Space—permeable soil | 38.1 (2.7) |
| Other Areas—impermeable soil | 91.5 (6.6) |
| Total Area (ha) | 1390.0 (100) |

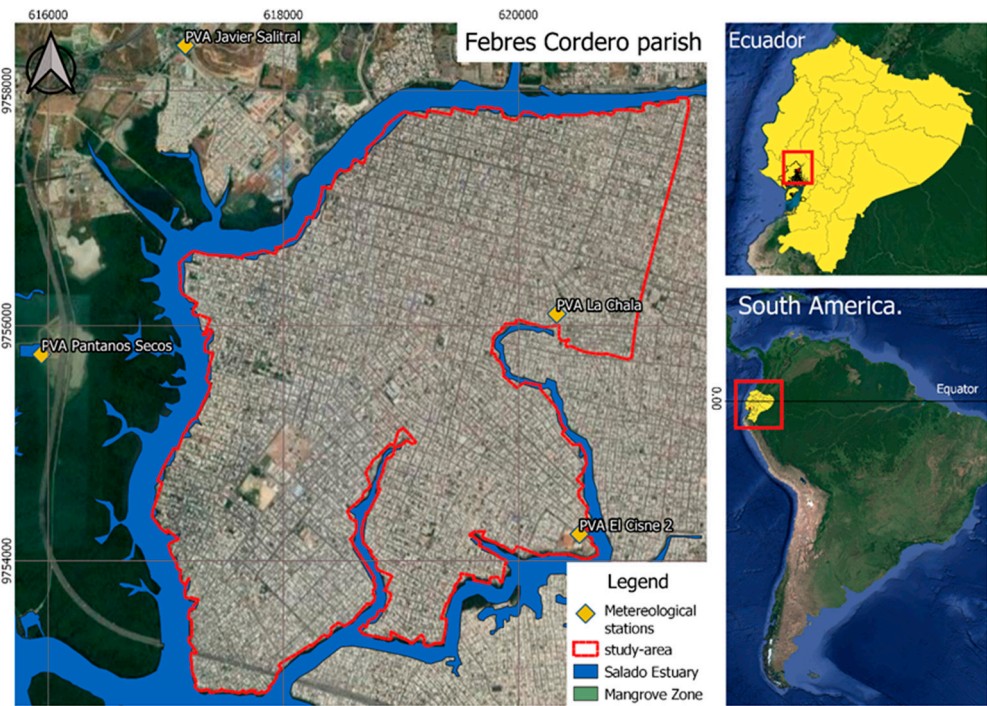

**Figure 1.** Location of the study area: Febres Cordero parish delimited in red line (**left**), the gray color represents the impermeable area cover by pavement (**left**) and geographical situation of the study area, Datum WGS84-17 South (**right**).

## 2.2. Hydrometeorological Information

Guayaquil has a tropical climate with a wet season from December to May with a total average rainfall of 1011.3 mm [19,20]. In the current study, four official meteorological stations were used (Table 2 and Figure 1) to analyze rainfall events due to the available 5 min fine temporal resolution. The information was previously filtered by CADS (Center for Water and Sustainable Development of ESPOL) and reviewed again with our reporting less than 2% of missing data at each station.

**Table 2.** Meteorological stations of the local area- DATUM: UTM WGS84-17S. Data provided by the CADS-ESPOL and INMHI.

| Official Station Code | Name | Longitude Coordinate | Latitude Coordinate | Time Series Length |
|---|---|---|---|---|
| M5157 | Pantanos Secos | 9755756.3 | 615942.7 | 2018–2019/20 months |
| M5165 | El Cisne 2 | 9754227.0 | 620518.9 | 2018–2019/20 months |
| M5156 | La Chala | 9756101.4 | 620325.8 | 2018–2019/20 months |
| M5164 | Javier Salitral | 9758385.4 | 617166.7 | 2018–2019/20 months |

To determinate the duration of the typical event, the data record including observations taken every 5 min at each station was used to identify storms, duration, and the volume of each of them. This data allowed for the analysis of rainfall distribution by normalizing accumulative rainfall (PI) and storm duration (TI) to total rainfall (PT) and total duration (TT), respectively [21,22]. The process was applied to each storm registered over the years 2018–2019 to produce normalized storms plots.

## 2.3. Storm Design for Different Periods of Time and Tidal Events

The guide for planning and determination of the maximum rainfall intensity, developed by the National Institute of Meteorology and Hydrology of Ecuador-INAMHI [23], was used. The report has the IDF Equations (1)–(3), which were developed from the Guayaquil International Airport-M0056 station, located at 10.8 km north of Febres Cordero.

From this guide, it is possible to build storm formation events associated with return periods of 2, 5, 10, 25, 50, and 100 years, and it was complemented with the typical duration event obtained by analyzing the finest events from the official stations selected (Table 2).

$$i = 135.7748 \times \frac{T^{0.2169}}{t^{0.3063}} \text{ for } 5 < t < 30 \text{ min} \tag{1}$$

$$i = 203.0259 \times \frac{T^{0.2169}}{t^{0.417068}} \text{ for } 30 < t < 120 \text{ min} \tag{2}$$

$$i = 1113.4537 \times \frac{T^{0.2169}}{t^{0.7779}} \text{ for } 120 < t < 1440 \text{ min} \tag{3}$$

where $i$ ($\frac{mm}{hr}$) is the intensity, $T$ (years) is the recurrence period, and $t$ (min) is the time of the duration of the event. Those equations helped to build the corresponding hyetographs. The rainfall hyetograph of design was built with the alternating block method [24] to characterize the events from different scenarios (i.e., rainfall and tidal events associated with different return periods).

Tidal data were obtained from the Tres Bocas station located 6 km away from the project, recording tidal waves in the Estero Salado to the north and west of FC parish. The magnitude of the tide was defined based on data from 2016–2019 of this official station. The equation, which was formulated by the official institution [25] and used to plot corresponding tidal events, is as follows:

$$Z = A \times sen \left[ \left( t - tp + \frac{1}{4f} \right) \times (2\pi f) \right] + (P - A) \tag{4}$$

where $A$ is the amplitude (m), $P$ is the tidal height (m), $tp$ is the peak rise time (h), $f$ is the frequency ($1/T$) (1/sec), and $Z$ is the magnitude estimated (m) of the tidal. The tidal events for different recurrence periods were obtained by a Gumbel distribution [26].

### 2.4. Hydrological Model to SWMM

The LIDs listed in Table 3 were adopted to reflect the assessed scenarios in SWMM. The Horton method for infiltration and the dynamic wave routing for solving one-dimensional Saint Venant equations for flow routing were applied. The Manning's equation to calculate the surface runoff and conduit flow were also followed [15]. Runoff, total inflow, and flooding were evaluated as metrics based on reduction rates of the peak flow on the whole system. The minimum reporting step time used to run the SWMM model was of 5 min (0.08 h).

**Table 3.** Pipe system, study area and proportions of LIDs facilities applied at FC.

| Item | Value |
|---|---|
| Subcatchments (units) | 1740 |
| Nodes (units) | 1716 |
| Outlets (units) | 134 |
| Conduits (units) | 1738 |
| Total Area (ha) | 1390 |
| LIDs (area, units) | Bioretention cells (19.5 ha), Rain Barrels (1740) |
| Total Area Lids (ha) | 19.51 |
| Proportion Lids (%) | 1.40 |

### 2.4.1. Subcatchment Delineation and Pipe System

Most of the drainage area is covered with pavement, and only a very small portion corresponds to green areas (Figure 1). Therefore, 98% of the drainage area is impervious (i.e., no infiltration can happen) and the rest is pervious (green areas). To quantify the runoff and the potential LIDs' implementation, the entire study area was subdivided into thousands of smaller subcatchments based on Thiessen polygons [27]. By using QGIS, 1740 polygons were delimitated. The criteria applied were determined on the basis of the

influence area of each manhole in correlation with the slope values of the digital elevation model (DEM) proportioned by the local government.

The evaluated stormwater system is composed of 120.3 km of pipes with diameters ranging from 0.16 m up to 2.25 m. The pipe system is connected by 1716 manholes (nodes), where the street runoff is collected. Finally, the system has 134 outlets discharging into the Estero Salado (Table 3) with more details shown in Appendix A, Figure A1, where the elevations throughout the area are also depicted.

### 2.4.2. Calibration and Estimation of Model Parameters

Critical hydrological and hydraulic parameters used in the SWMM simulation include the width of the subcatchment area, Manning's coefficients, and subsidence depth. Other parameters such as pipe length and diameter, node elevation, surface area, and ratio of the impermeable area were obtained from the municipal stormwater master plan [24]. Soil type was determined preliminarily by the current situation of the study area (photographs), recommendations of the SWMM manual [15], and related literature pertinent to the zone [19,28].

Due to a high number of polygons in the study area, the initial set-up was performed by applying the swmmr-R package [29] (the script used can be found within the Supplementary Materials "Scrip S1_swmm.r"). Different values of parameters utilized (Tables 4 and 5) were estimated through a combination of field data, literature review, and model defaults according to the recommendation of Rossman and other works [15,30,31].

**Table 4.** Initial input parameters to model Febres Cordero rainfall system.

| SWMM Parameters | Description (Units) | Estimate Range | Reference | Calibrated Value |
|---|---|---|---|---|
| Flow Width | Width of overland sheet flow (m) | 0.2 to $5 \times \sqrt{area}$ | [32] | $\sqrt{area}$ |
| N Imperv | Mannig's Roughness of impervious areas | 0.01 to 0.02 | [15] | 0.015 |
| N perv | Manning's Roughness for pervious areas (for grass) | 0.040 to 0.140 | [15] | 0.14 |
| DStore Imperv | Depth of depression storage on impervious areas (mm) | 1.3 to 2.5 | [15] | 1.5 |
| Dstore Perv | Depth of depression storage on pervious areas(mm) | 2.5 to 7.5 | [15] | 4 |
| MaxRate | Hortons Maximum Infiltration Rate (mm.hr) | 30 to 200 | [33,34] | 125 |
| MinRate | Hortons Minimum Infiltration Rate (mm/h) | 0.1 to 20 | [34] | 4 |

**Table 5.** Parameters to configure Bio-retention cells.

| LID Layer | Parameters | Bio-Retention Cell |
|---|---|---|
| Surface | Berm Height (mm) | 100 |
| | Vegetation Volume (fraction) | 0.1 |
| | Surface Roughness (Manning's n) | 0.1 |
| | Surface Slope (percent) | 1 |
| Soil Media | Thickness (mm) | 400 |
| | Porosity (volume fraction) | 0.5 |
| | Field Capacity (volume fraction) | 0.2 |
| | Wilting Point (volume fraction) | 0.09 |
| | Conductivity (mm/h) | 250 |
| | Conductivity Slope (percent) | 10 |
| | Suction Head (mm) | 60 |
| STORAGE | Thickness (mm) | 500 |
| | Void Ratio (voids/solids) | 0.75 |
| | Seepage Rate (mm/h) | 600 |
| | Barrel Height (mm) | NA |

NA = Not Applicable.

Along the Febres Cordero pipe network, 8 of the 134 outlets are usually flooded according to the SWMM. To set up border conditions, 4 points were verified with photographs, due to the municipality of Guayaquil not having up-to-date information from monitoring of the pipe system. A visual inspection to verify the level of the water was performed (see Appendix A, Figure A2).

*2.5. Low-Impact-Development Strategies*

The bio-retention cells, shown by the conceptual model in Figure 2, were implemented on the streets of 1.5 m in width and follow street slopes of around 3% according to the DEM in the study area. Table 5 shows the input parameters for bio-retention cells. Regarding rain barrels, those parameters do not apply.

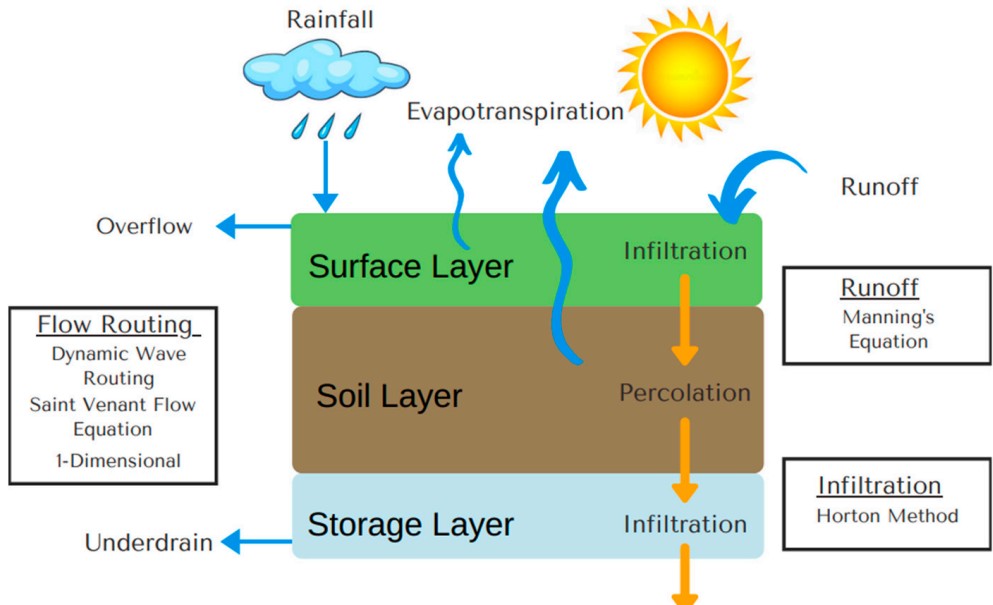

**Figure 2.** Conceptual hydrological process for bio-retention cells in Stormwater Management Model (SWMM).

Due to a lack of information at selected meteorological stations, the evapotranspiration parameter was estimated exclusively based on another station named the University of Guayaquil station, which is located 5 km to the north of the study area. According to [35], the evapotranspiration obtained by the Thornthwaite method [36] was about 4.59 mm/d. This value was applied to the model, especially for bioretention cell design (see Appendix A, Figure A3).

Due to the street geometry in the study area, it is possible to apply LID strategies as green streets, mainly in streets that are more than 8 m in width. Appendix A, Figure A4 shows a conceptual illustration of this solution applied to the study area. More details can be found in the Supplementary Materials, Figure S1. In the analysis, street geometry and traffic density were considered, utilizing information from Google Maps with its live traffic feature [37]. Around 37 km of streets were selected during peak hours, generally between 12:00 and 13:00 at midday and during the afternoon from 16:00 and 19:00, as shown in Figure 3.

*2.6. Validation*

The validation of the model is currently unachievable due to non-existent monitoring of the pipe system in Guayaquil. However, to reduce the uncertainty of the model results, the boundary conditions and sensitive analysis of the Manning coefficient in this article were set in accordance with site photographs (see Appendix A, Figures A2 and A5). Calibrated results in this article will serve to grow awareness amongst municipal authorities in regard

to the importance of keeping flood records to improve modelling results for decision-making processes.

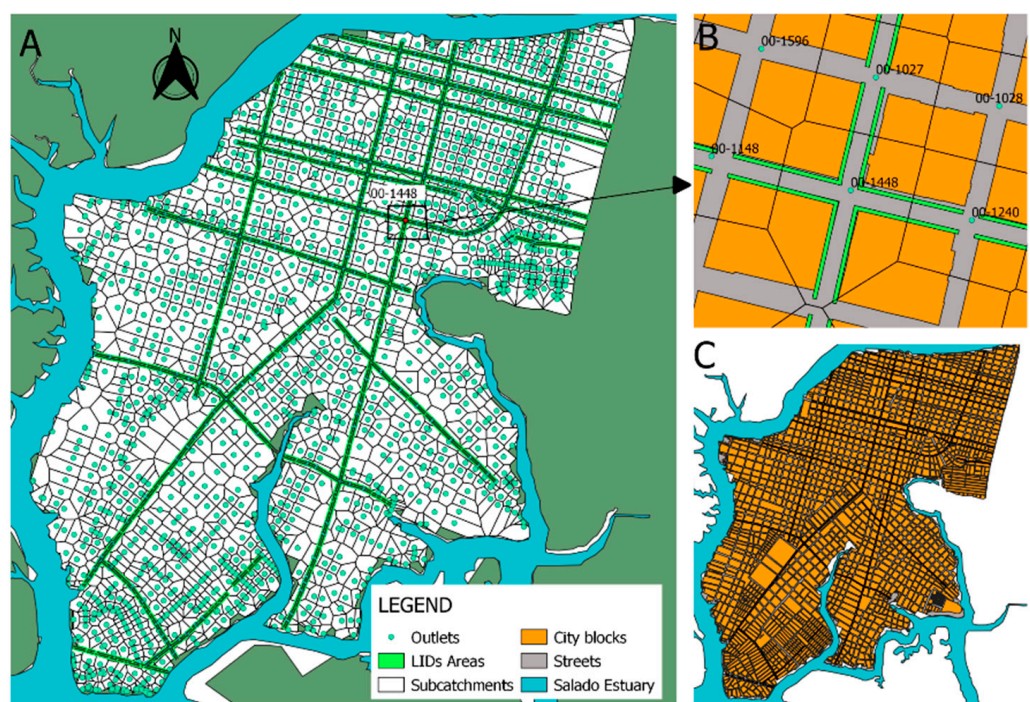

**Figure 3.** Subcatchment delineation and pipe system for the Febres Cordero parish: (**A**) Subcatchments based on Thiessen polygons in the study area, (**B**) View of a subcatchment composed of area LIDs, streets and city blocks with impervious surface, (**C**) Discretization of the study area to locate 1740 rain barrels of 55 gallons as LID strategy, one for each block.

## 3. Results

### 3.1. The Duration of the Typical Event at Febres Cordero

Around 150 events with a temporal resolution of 5 min were analyzed from the four meteorological stations available. 78 events with a duration greater than 20 min are normalized and plotted on Figure 4. The typical synthetic storm is indicated by the red line which was obtained based on the density of the curves (i.e., it represents the median of all of them) [21]. This graph shows us that the duration of the typical event that happens in FC is about 200 min.

### 3.2. Rainfall Events and Tidal Waves

With a duration of the typical rainfall event of 200 min, the design events are as follows:, the rainfall volume is 69.69 mm for a 2-year event, 85.35 mm for a 5-year event, and 99.2 mm of rainfall for a 10-year event (Figure 5a). For extreme events, over 25-, 50-, and 100-year return periods, estimated rainfall volumes are 187.6 mm, 218.1 mm, and 253.4 mm, respectively. All of them are built with a rainfall duration of 24 h. The hyetographs in Figure 5b represent the three scenarios.

The levels for extreme tidal events are of 2, 2.1, 2.15, 2.22, 2.4, and 2.6 m for the events associated with return periods of 2, 5, 10, 25, 50, and 100 years, respectively (Figure 6).

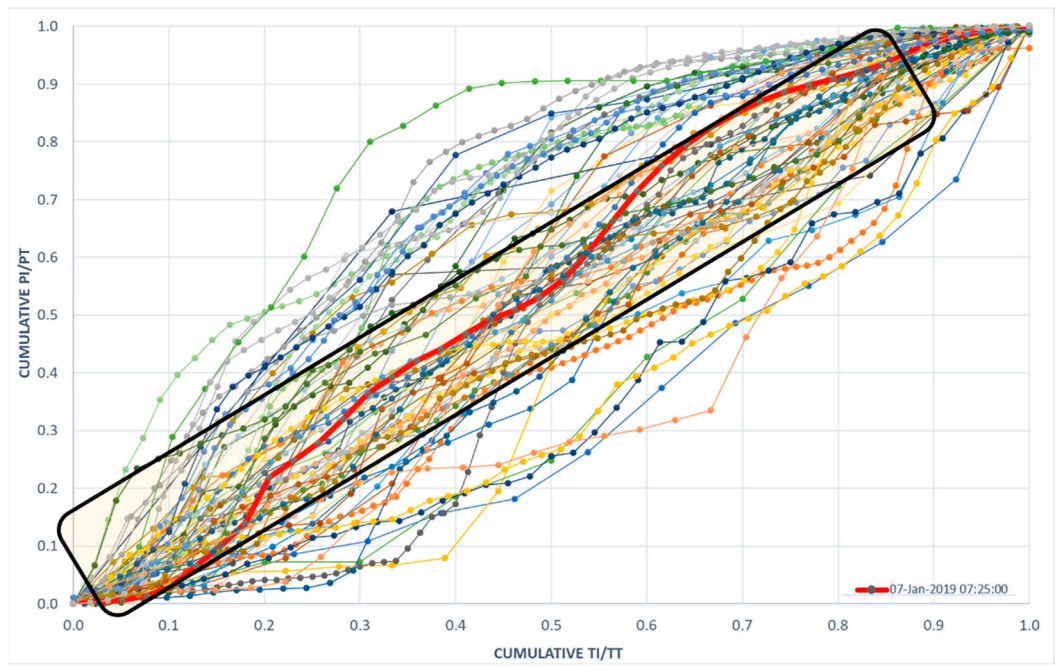

**Figure 4.** 78 normalized storms plotted from March 2018 to July 2019. The typical event is the storm which occurred on January 07 of 2019 (red line) and was registered by the "Pantanos Secos station". It started at 7:25 a.m. and finished at 10:45 a.m. Normalization processes considered accumulative rainfall (PI) and storm duration (TI) to total rainfall (PT) and total duration (TT) respectively.

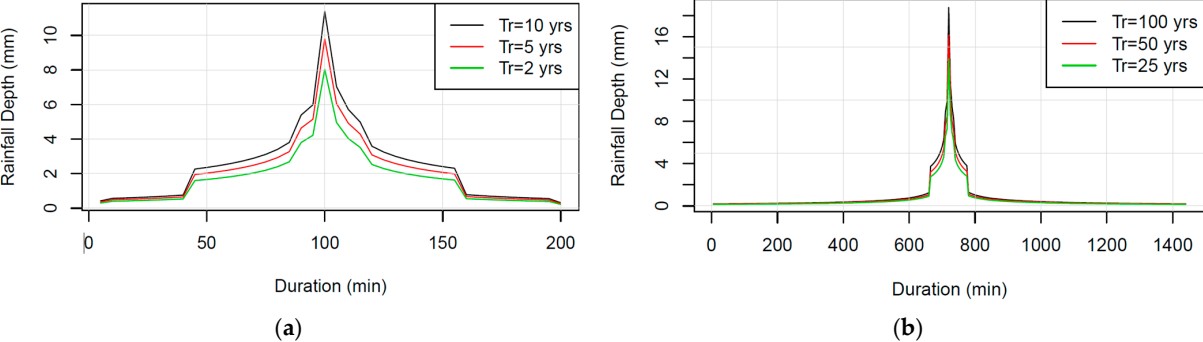

**Figure 5.** (**a**) Alternating Block Method for a short rainfall design; (**b**) Alternative Block Method for a long rainfall design.

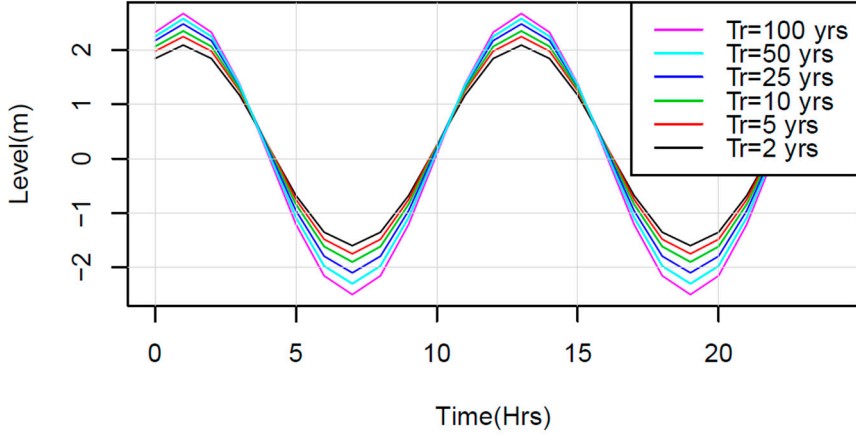

**Figure 6.** Design tidal events for Febres Cordero parish.

### 3.3. Drainage Pipe System and LID Strategies Modelled

3.3.1. Overflow Nodes Simulation Analysis

In terms of node flooding, a node is considered as overflowing when the flooding time is more than 30 min. The simulated system shows that in less than 30 min, the water should be drained through the pipes. The flooding time range is mainly centered within the typical rainfall duration designs shown in Table 6.

**Table 6.** Simulation analysis: TD: Traditional Development mode; LID: Low impact development mode.

| Rain Events (Tr) | Rainfall Duration (HH:MM) | Number of Overflow Nodes | | Duration Range (h) | | Max Flow Rate (m³/s) | |
|---|---|---|---|---|---|---|---|
| | | TD | LID | TD | LID | TD | LID |
| 2 | | 428 | 311 | 0.08–4.48 | 0.08–4.31 | 8.78 | 7.99 |
| 5 | 3:20 | 581 | 448 | 0.08–4.63 | 0.08–4.39 | 9.44 | 9.23 |
| 10 | | 656 | 568 | 0.08–4.66 | 0.08–4.45 | 9.72 | 9.46 |
| 25 | | 773 | 741 | 0.08–23.21 | 0.08–22.42 | 9.79 | 9.75 |
| 50 | 24:00 | 849 | 826 | 0.08–23.28 | 0.08–22.60 | 10.21 | 10.08 |
| 100 | | 924 | 912 | 0.08–23.35 | 0.08–22.75 | 10.26 | 10.26 |

For a 2-year design event, a 27% reduction in flooding nodes is achieved with LIDs in comparison with a traditional approach. For 5-year and 10-year events, flood reduction is 23% and approximately 13%, respectively. For extreme periods with a duration of 24 h, flood reduction is around 4% for a 25-year event, 3% for a 50-year event, and 1% for a 100-year event.

3.3.2. Total and Peak Runoff

The whole urban catchment area is comprised of 97.3% impermeable soil. This means that almost the total inflow at the system is due to runoff from impervious areas. The evaporation loss is small at around 1.3% of the total rainfall. The infiltration loss is around 0.58% from the total rainfall, which means that those parameters are considered negligible during the simulation. The achieved reduction rates from the LID approach are significantly higher than from traditional drainage systems, as shown in Table 7.

**Table 7.** Total runoff analysis in study area under the TD: Traditional Development mode; LID: Low impact development mode.

| Rain Events (Tr) | Duration Rainfall (HH:MM) | Rainfall (mm) | Total Inflow (m³) | Total Runoff of TD (m³) | Total Runoff of LID (m³) | Reduction Rates of TD (%) | Reduction Rates of LIDS (%) |
|---|---|---|---|---|---|---|---|
| 2 | | 70 | 930,481.80 | 921,858.60 | 751,766.00 | 1 | 20 |
| 5 | 3:20 | 85 | 1,142,908.13 | 1,130,111.38 | 949,918.38 | 1 | 17 |
| 10 | | 99 | 1,338,533.25 | 1,317,601.38 | 1,130,598.50 | 2 | 16 |
| 25 | | 188 | 2,515,853.00 | 2,436,473.50 | 2,033,450.25 | 3 | 19 |
| 50 | 24:00 | 218 | 2,938,373.50 | 2,845,306.00 | 2,412,733.50 | 3 | 18 |
| 100 | | 253 | 3,442,607.25 | 3,320,914.00 | 2,844,423.00 | 4 | 17 |

The total runoff reduction for 2-,5-, and 10- year events with short duration (200 min) are improved by 19%, 16%, and 14% under the LID mode, respectively, compared to the TD mode. For long duration rainfall events of 25-, 50-, and 100-year recurrence periods, reductions are improved by 16%, 15%, and 13%, respectively, under the LID mode compared to the TD mode.

The peak runoff of the baseline scenarios (i.e., TD mode) and contrasted with the LID mode has a max ratio of reduction of 21% for the 2-year recurrence period (279 m³/s and 220.7 m³/s, respectively), 18% for the 5-year recurrence period (349.7 m³/s and 289.4 m³/s, respectively), and 17% for the 10-year recurrence period (414.3 m³/s and 345.4 m³/s, respectively). Hyetographs are shown in Appendix A and Figure A6a–c. For extreme events,

a reduction of 15% for the 25-year recurrence period (527.6 m³/s and 442.1 m³/s, respectively), 14% for the 50-year recurrence period (611.85 m³/s and 523.7 m³/s, respectively), and 14% for 100-years recurrence period (722.6 m³/s and 620.6 m³/s, respectively) were observed. Hyetographs are shown in Appendix A, Figure A6d,e.

### 3.3.3. Flooding and Performance of LID Strategies

The flooding system peak reductions of the TD mode contrasted with the LID mode for short term events are 26% for the 2-year recurrence period (209.2 m³/s and 154.6 m³/s, respectively), 21% for the 5-year recurrence period (293 m³/s and 230 m³/s, respectively), and 19% for the 10-recurrence period (361.9 m³/s and 293.4 m³/s, respectively). Hyetographs are shown in Figure 7a–c. The reductions for extreme events are 12% for the 25-year recurrence period (433.9 m³/s and 383.2 m³/s, respectively), 15% for the 50-year recurrence period (548.9 m³/s and 468.4 m³/s, respectively), and 13% for the 100-year recurrence period (652.6 m³/s, and 566.4 m³/s, respectively). Hyetographs are shown in Figure 7d–f (for more details of the physical implementation of LID strategies, refer to the video "Video S1_Resilients Streets" found within the Supplementary Materials section).

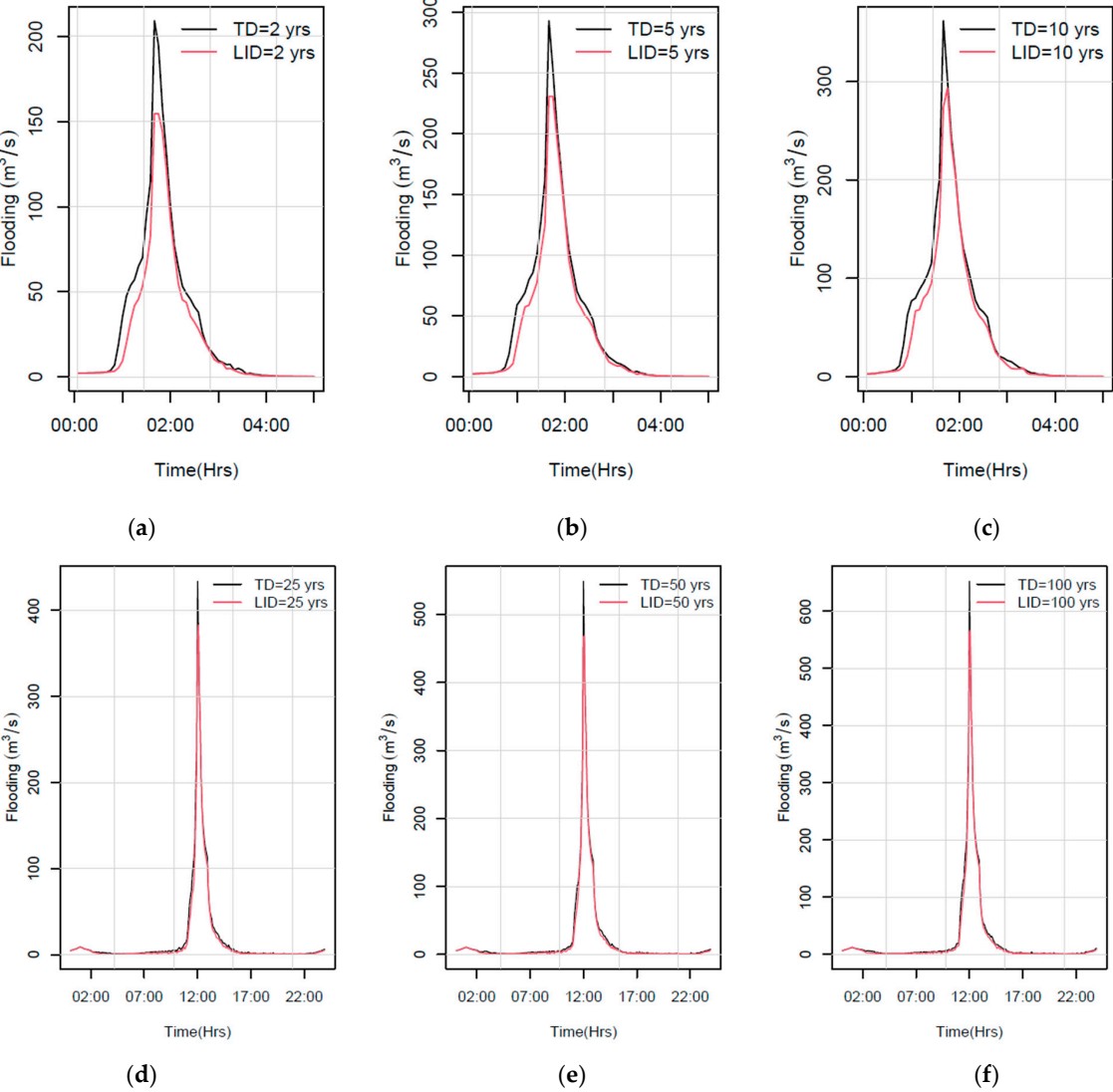

**Figure 7.** Peak reduction of the flooding system of events with a typical duration of 200 min: (**a**) 2-year recurrence period; (**b**) 5-year recurrence period; (**c**) 10-year recurrence period. Peak reduction of the flooding system of events with a typical duration of 200 min: (**d**) 25-year recurrence period; (**e**) 50-years recurrence period; (**f**) 100-year recurrence period.

The flow continuity error calculated by SWMM was less than 3% for surface runoff as well as for flow routing (i.e., minor inconsistent pipe connections as low slopes, or negative values). The general results of the whole system are presented in Table 8. Here, values from overflow, peak flooding, and peak runoff are presented. Hyetographs related to performance of runoff are shown in Appendix A, Figure A6a–f.

**Table 8.** Performance of the system on TD and LID mode for various recurrence period.

| Return Period (Years) | Overflow Nodes | | Rate of Reduction (%) | Max Rate Node Overflow (m³/s) | | Rate of Reduction (%) | Peak Flooding System (m³/s) | | Rate of Reduction (%) | Peak Runoff (m³/s) | | Rate of Reduction (%) |
|---|---|---|---|---|---|---|---|---|---|---|---|---|
| | TD | LID | | TD | LID | | TD | LID | | TD | LID | |
| 2 | 428 | 311 | 27 | 8.8 | 8.1 | 7 | 206 | 160 | 22 | 279 | 221 | 21 |
| 5 | 581 | 448 | 23 | 9.4 | 9.2 | 2 | 293 | 231 | 21 | 350 | 286 | 18 |
| 10 | 656 | 568 | 13 | 9.7 | 9.5 | 3 | 362 | 293 | 19 | 414 | 345 | 17 |
| 25 | 773 | 741 | 4 | 9.8 | 9.8 | 0 | 434 | 383 | 12 | 518 | 442 | 15 |
| 50 | 849 | 826 | 3 | 10.2 | 10.1 | 1 | 549 | 468 | 15 | 612 | 524 | 14 |
| 100 | 924 | 912 | 1 | 10.3 | 10.3 | 0 | 653 | 566 | 13 | 723 | 621 | 14 |

## 4. Discussion

### 4.1. Modeling Typical Events and Extreme Scenarios

The duration of the event, generally originating from a unimodal regimen and convective form of precipitation as found by Ballari in [20], also seems to coincide with the time identified on a study developed in the city of Duran [38], where the study site was 16 km from the Febres Cordero parish. As well, the hydrological data were obtained by the official meteorological station known as *Divino Niño* operated by INAMHI.

The effect of the Pacific Ocean increases the variability of the events [39]. According to [19], the weather and climate change are altering the patterns and potentially the principal drivers of the climate at all scales. Therefore, the extreme events obtained through the official reports from INAMHI, which have been widely used [40–42], represent probabilistic scenarios that are not far away from the reality of this study area. Moreover, studies such as [43] have characterized the influence of climate drivers such as El NIÑO and even the rise in sea levels, scenarios which were considered in the present study as extreme events.

Previous research applying SWMM has pointed out the most relevant variables for model performance such as estimation of roughness coefficients, energy losses along pipes, overland flow exchange, and DEM resolution [44,45]. Here, the local knowledge (from people who live in the area) with regard to the use of the different structures and the individual size of private properties (to quantify the impermeable area) was used to recalibrate the variables of the model. Because the missing information on the pipe invert levels was about 3% for the study area, photos taken by the authors and a DEM were used for this purpose.

### 4.2. Low Impact Strategies and Future Opportunities

Bioretention cells have the advantage of improving water quality as natural filtration for particulate matter. A submerged (anoxic) zone combined with carbon source are ideal for coastal cities, where the benefits are amplified, thereby enhancing the removal of nutrients and metals in the bioretention systems [17,46,47]. Moreover, some research has indicated that bioretention cells are more cost-effective for peak flow reduction than other green solutions, as well as helping to mitigate heat islands formed by the excessive use of pavement [15,48]. However, activities such as weltering, sediment removal, vegetation management, mulching, and pest control of bioretention systems could represent higher operation and maintenance costs if there is not a well-defined long-term plan. The other solutions proposed were the rain barrels, which can be used to reduce operation and maintenance activities (e.g., runoff retained in barrels could be used to maintain bioretention cells with the help of the local population) [49].

Blue and green infrastructure around the world has been successfully applied to mitigate flooding [50–52]. In Ecuador, despite having the legal framework of a pioneering constitution that recognizes the rights of nature, and along with efforts to explore new opportunities based on nature-based solutions, the mere implementation of low impact developments is not sufficient [53,54], mainly due to the lack of planning regulations at the local level (for example in the Decentralized Autonomous Governments—GADs or municipalities), where water management decisions and the application of a holistic approach are not appropriate. On the other hand, social factors also play a key role, where a quicker and more efficient method for the sustainable adaptation of these solutions is through the participation and empowerment of citizens, particularly within the Febres Cordero parish [55–58], thus creating synergies that would help resolve existing trade-offs between the Sustainable Development Goals (SDGs) [59,60]. Particularly, SDGs 6,11,14, and 15, which are related to the sustainable management of water, show us that the implementation of this kind of solution (LIDS) is not an option, but is a necessity.

*4.3. Uncertainty of the Modeling*

The selection of the meteorological station for the study site was determined based on the availability, proximity, and quality of the data that each station provided. The missing data were less than 2%, and the storms were extracted without any problems for the data. Some events that accumulated over 2000 mm in less than one hour were deleted due to their representation as outlier events that rarely occurred and were interpreted as incorrect functions of the pluviometer.

Waterlogging normally occurs in low-lying areas, where the pipe system exceeds the upper limit of hydraulic capacity. Most of the outfall nodes are located near sea level (0 m a.s.l.), and some of them at −1.67 m a.s.l. This implies that all pipes connected to most of the outfall will potentially be flooded when the high tide peak reaches 2.09 m a.s.l. To verify if the model corresponds to reality, several photos were collected for verification purposes (see Appendix A, Figure A2).

The calibration of Manning's coefficient for concrete pipes that were deployed more than 20 years ago showed coherence with the reference water depths in photographic records of frequently flooded locations ($r^2 > 0.85$). To reduce the uncertainty of the results, street geometry (e.g., street elevations, road-curb size, and curb height) was used as a reference point on photographs to check the levels that the model reported. Unfortunately, municipalities do not collect systematically and reliable information about pipes, channels, and flood storage areas. This article also shows the necessity of developing an urban data platform linked to a geographic information system with hydrological and topographical information to inform better decision-making.

## 5. Conclusions

The model showed the estimation of how the Febres Cordero pipe system works. Despite data limitations and considering normal and extreme design events in the near future, the performance of the pipe system showed that even in less intense scenarios (i.e., 2-year design events with 200 min of duration), the current design is not enough to convey higher volumes as evidenced by the system, which demonstrated flooding. This shows how vulnerable the FC is to rain and tidal events.

The area where LIDs was implemented represents 1.4% (19.5 ha) of the whole study area and was able to reduce 20% of the runoff for short events (200 min), with an upgrade of 19% over TD. For extreme events (24 h duration), the reduction rate was 19% with a maximum upgrade of 16% in comparison to the TD. This shows the efficient improvement of the pipe system performance due to LID application only in a small portion of the Febres Cordero. This highlights the potential applicability of green and blue infrastructure in places that have been struggling with flooding problems under existing negligent conditions.

Anthropogenic factors such as a high increase of impervious surface and waste accumulation are expected to increase flood conditions (i.e., height and duration of events).

The rapid increase of informally populated areas which previously were grasslands and mangroves are contributing to this growing problem. This study is also a reminder that a holistic approach is required for urban planning, where authorities and stakeholders should collaborate with each other. In addition, increasing city resilience requires an understanding of the socio-ecological drivers of urban flood hazard and risk, continuous monitoring, and spatial data collection of water stages and velocities that are complemented by data collection devices of the latest generation (but are not expensive). In this context, security cameras could be used as a real-time monitoring system that would generate information that can improve hydrometeorological modeling for the development of early warning systems for flood events at the city level, which will reduce social and economic losses.

**Supplementary Materials:** The following supporting information can be downloaded at: https://espolec-my.sharepoint.com/:f:/g/personal/flquichi_espol_edu_ec/EhlJqzs-VvtEvjfpSSCFW0kBhN5BI0KAItHAi4s1yeG7Cg?e=biB0hl, (accessed on 7 June 2021), Figure S1: Resilient streets sketch; Video S1: Resilients Streets; Script S1: swmmr.R.

**Author Contributions:** Conceptualization, F.Q.-M.; methodology, F.Q.-M.; software, F.Q.-M.; formal analysis, F.Q.-M., D.M., P.Q.-M. and L.J.; investigation, F.Q.-M.; data curation, F.Q.-M.; writing—original draft preparation, F.Q.-M. and D.M.; writing—review and editing, F.Q.-M., D.M., P.Q.-M. and L.J.; visualization, F.Q.-M. and L.J.; supervision, D.M. and P.Q.-M.; funding acquisition, P.Q.-M. All authors have read and agreed to the published version of the manuscript.

**Funding:** This research received no external funding, and the APC was funded by the Vicerrectorado de Investigación of the Universidad de Cuenca.

**Acknowledgments:** The authors acknowledge the participation of "Centro de Agua Y Desarrollo Sustentable" CADS as institution of investigation of ESPOL who supported the access to the official information of the meteorological stations. Finally, we would like to thank Gregory Gedeon for text revision.

**Conflicts of Interest:** The authors declare no conflict of interest.

**Appendix A**

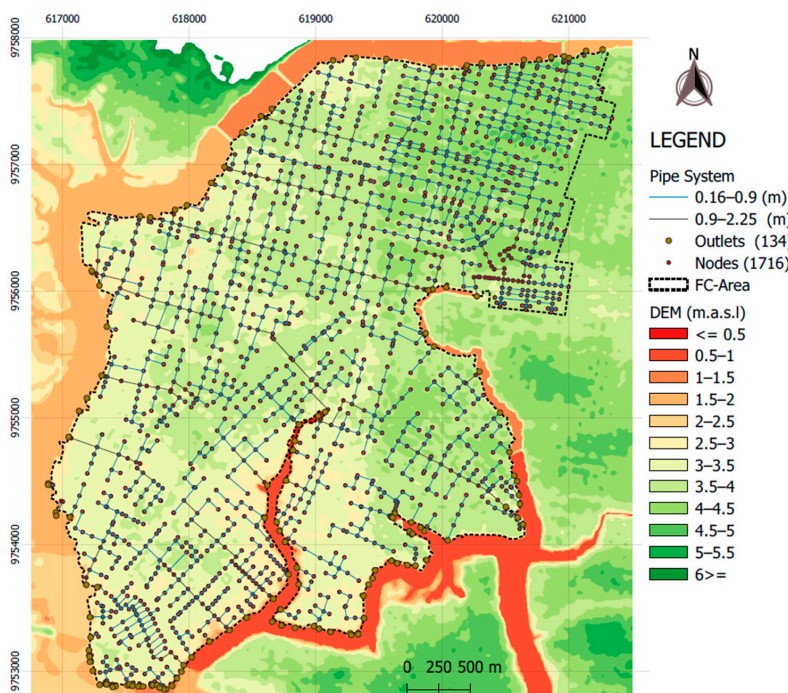

**Figure A1.** Digital elevation model and stormwater system composed by 120.3 km of longitude in the study area.

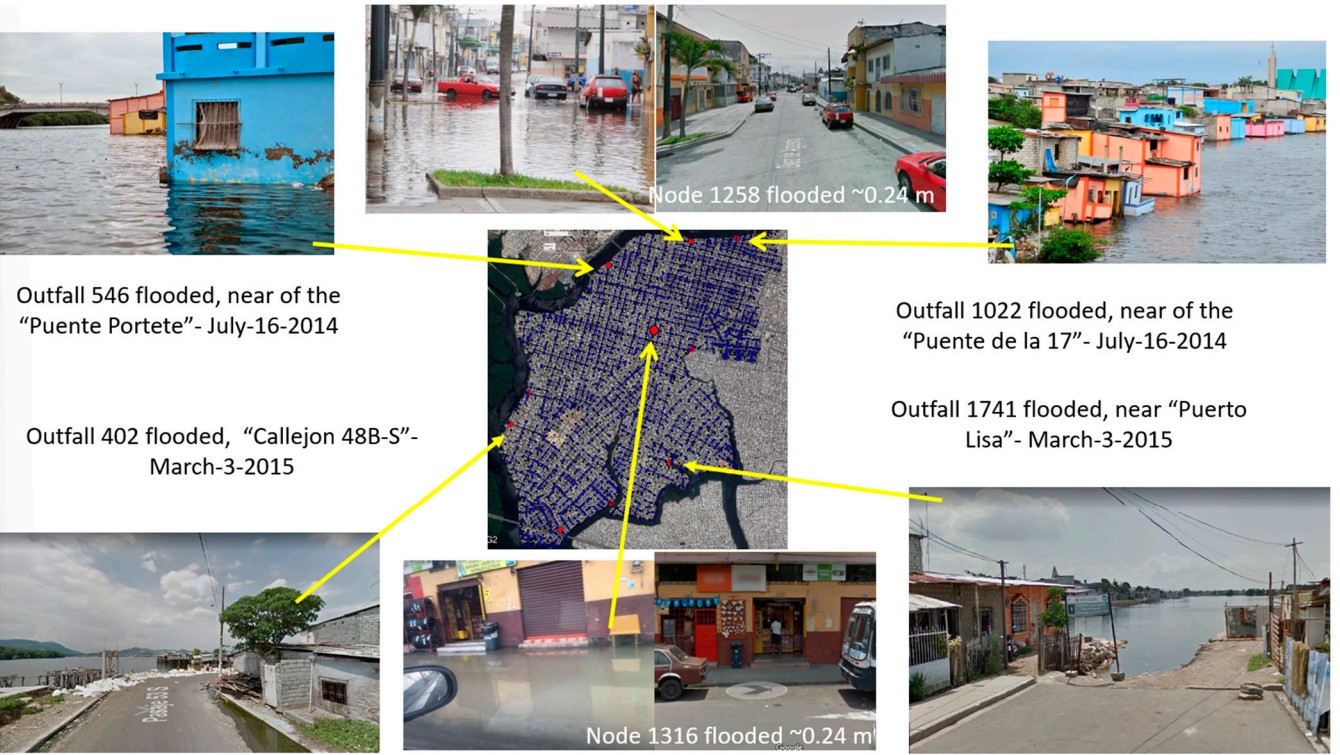

**Figure A2.** Calibration of boundary conditions with levels checked with photographs. Photos taken by the authors.

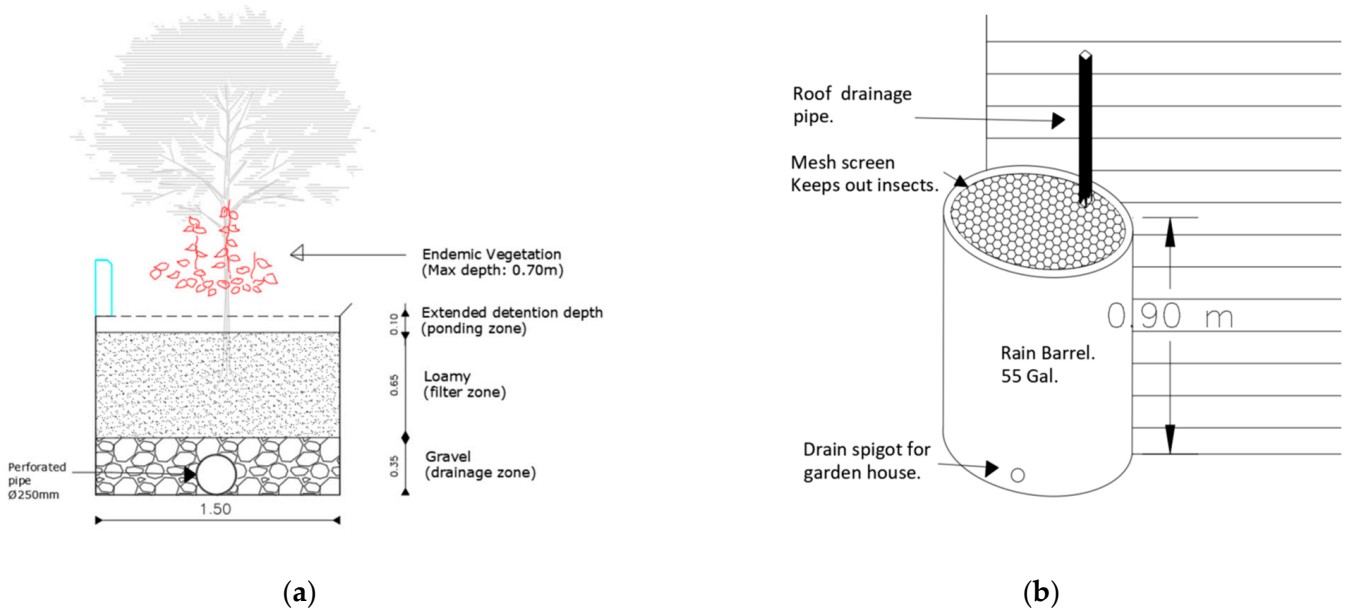

(**a**)

(**b**)

**Figure A3.** Example of LID applied of the study area: (**a**) Bioretention cells, and (**b**) Rain barrels.

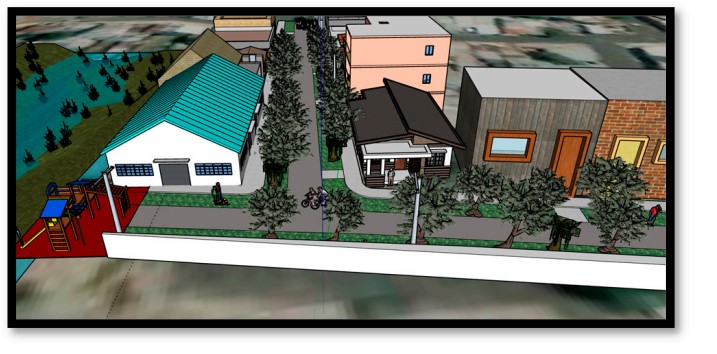
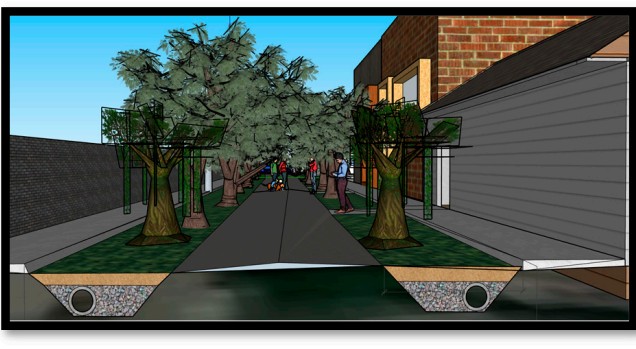

(**a**)  (**b**)

**Figure A4.** Conceptual illustration of green streets for the Febres Cordero parish: overview illustration of green streets (**a**) and transversal section of LID applied (**b**).

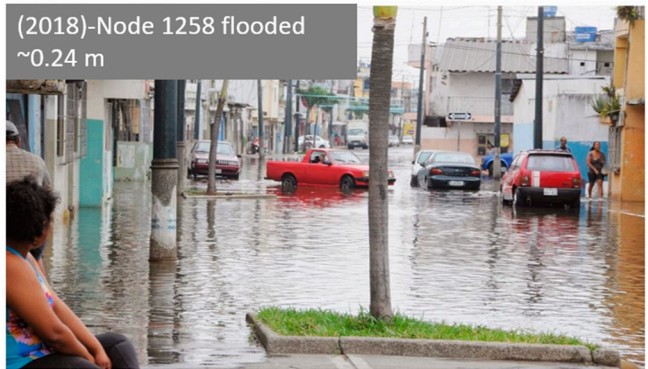
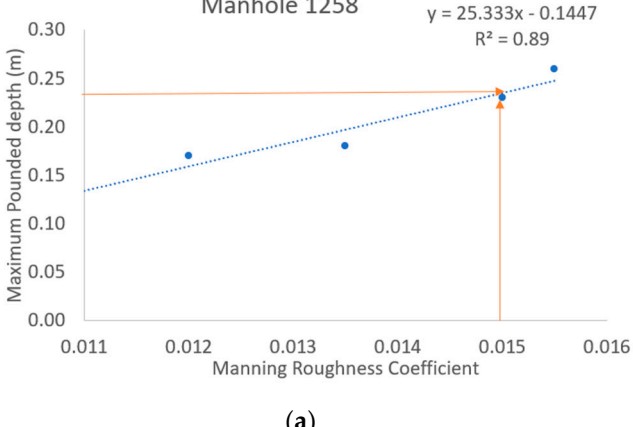
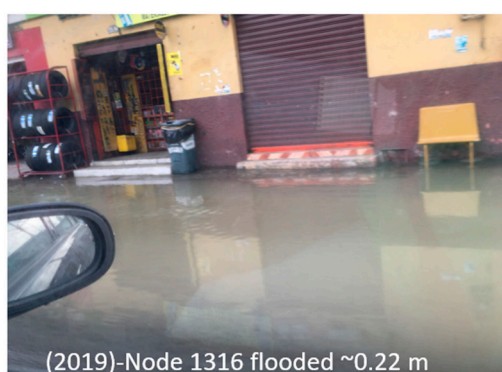
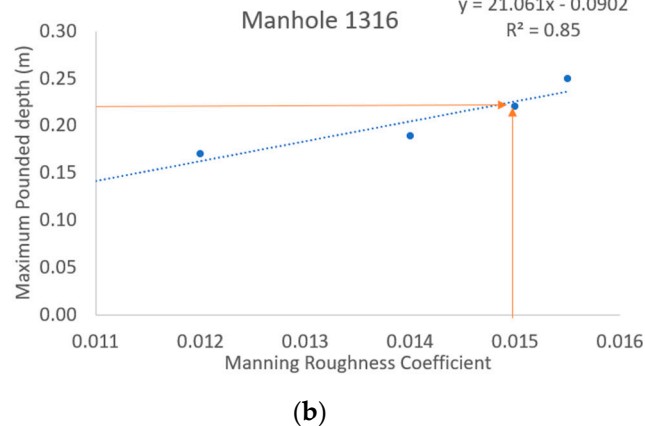

(**a**)  (**b**)

**Figure A5.** Sensitive analyzed to the manning coefficient according to the flooded levels registered on site. On each point, the model ran four times to obtain each refence Manning's value and level of flooding. (**a**) Point located between the 10 de Agosto Street and Otavalo street. (**b**) Potin located between Rafael Garcia Goyena street and Avenida 43.

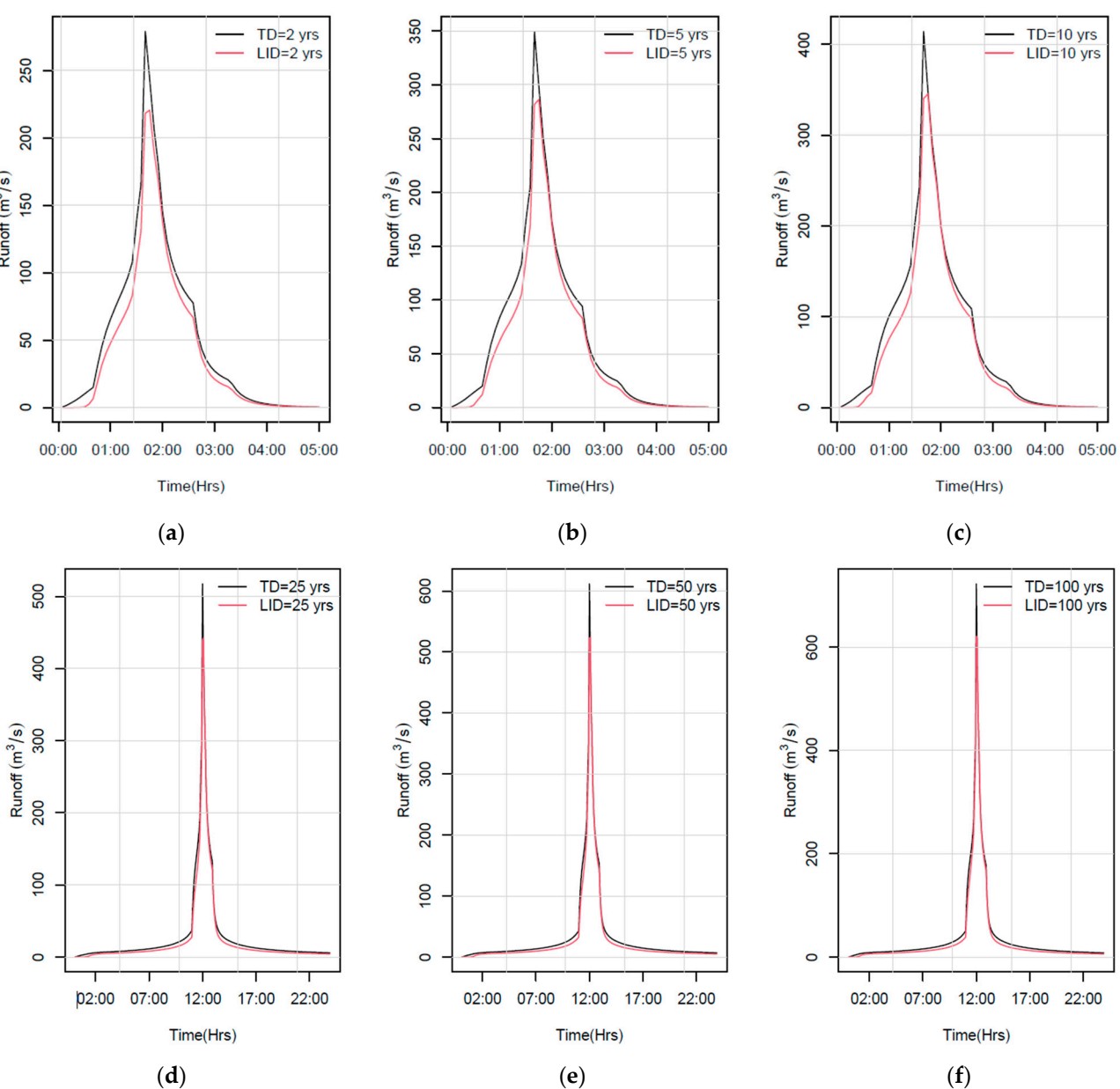

**Figure A6.** Peak reduction of the runoff on events with a typical duration of 200 min: (**a**) 2-year recurrence period; (**b**) 5-years recurrence period; (**c**) 10-year recurrence period. For extreme events with duration of 24 h (**d**) 25-year recurrence period; (**e**) 50-years recurrence period; (**f**) 100-year recurrence period.

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
