# Peer review of "Influence of Low-Impact Development in Flood Control: A Case Study of the Febres Cordero Stormwater System of Guayaquil (Ecuador)"

_sustainability, doi:10.3390/su14127109_

Round 1

Reviewer 1 Report

The manuscript consists of evaluating the implementation of two Low Impact Development Strategies (LIDS) – green streets and rain barrels – as a solution to reduce flooding in a coastal zone in Ecuador as a case study using the Stormwater Management Model (SWMM).

The study is considered a case study without any new novelty in methodology. In my opinion, the methodology was not clear, which had an effect on the scientific value of the study. Also, I did not find any new novelties through the reading. I think the authors have done a lot of work, but you get the feeling that the form of the presentation is made for a scientific report. Another important concern of the study is not taking into consideration the impact of the expected climatic changes on the region for the studied future periods. All modern hydrological studies cannot be adopted without showing the impact of climatic changes on it.

In general, I do not see any new point or a significant contribution to this paper in its current form and unfortunately, I will recommend rejecting the article.

Author Response

We have added our response to the document attached below.

Reviewer 2 Report

The paper is well organized in a logical way. Although I am not a language expert, The manuscript needs to check for any grammar, spelling, and punctuation errors. I have a comment, the Uncertainty in the results should be added and discussed in the discussion section. Also, the conclusion is very simple and it should be extended.

some small errors:

 I prefer to use present tense for describing the result since results show the finding now, and not in past. Line 254, and 269 “,” are necessary before “respectively”. In line 276 “The evaporation loss is small, around 1.3% from the total rainfall” using “,” is extra. In line 287 should use “,” before “and”. They are a few examples of errors I have seen. 

Good luck.

Author Response

We add our answers on a document attached below. 

Reviewer 3 Report

Influence of low impact development in flood control. A case study of Febres, Cordero stormwater system at Guayaquil City (Ecuador). 

This is. And interesting research work. However, However, there are some issues to be sorted before it goes for publication. 

Abstract 

I can see several typos. Please correct these typos. For example, the first line of the abstract has a mayor problems. It should be major problems. Likewise, there are some other typos throughout the manuscript. Please correct these typos. 

The second sentence of the abstract. What do you mean by quality on land use? 

4th sentence of the abstract. The authors have started the sentence using “through the stormwater management model”. You have to correct your English. 

The authors haven't showcased the research gap in the abstract. Please tell why you need this research work. 

Introduction. 

Orders have presented 3 objectives in this research. However, it would be better to state these objectives just after the research gap. Even in the introduction, the authors have not successfully showcased the research gap.  

Materials and methods. 

Table 2. How do you define these? North coordinate and east coordinate. You have 6 digits. 

What is the resolution of the data? Are these daily? Or monthly rainfall. Are there any missing data? Format the equations correctly. 

Results. 

Explain the Figure 4. It's very unclear. How did you validate your results? Hydraulic models can be developed by any sophisticated models. But validation in the real world is very important. 

Discussion. 

I personally like results and discussion to be in one section. However, authors can decide it. Nevertheless, the discussion is very weak.  

Conclusions 

What you have presented is rather a summary. Please state the real conclusions of your research work. And discuss them with your future work. I still couldn't find the real contribution of your research work. 

References. 

A good number of references are cited. Therefore, It should be fine. 

Author Response

We have added our answers to the document attached below.

Round 2

Reviewer 3 Report

Revisions are acknowledged. 

However, I still have this issue in Table 2. 

Can you please type your latitude and longitude coordinates in Google Earth and find the place that you are referring to?

I rather tried to find, but couldn't. If you are referring a different coordinate system, please could you reveal that?

Usually coordinates are given in this format;

El Cisne - 2.21800S, -79.93030W

La Chala - 2.20780S, -79.91470W

Please can you explain this for clarity?